

# Simulating the effects of climate change across the geographical distribution of two medicinal plants in the genus *Nardostachys*

Junjun Li[1], Jie Wu[2], Kezhong Peng[2], Gang Fan[1], Haiqing Yu[2], Wenguo Wang[3] and Yang He[1]

[1] College of Medical Technology, Chengdu University of Traditional Chinese Medicine, Chengdu, China
[2] Ganzi Prefecture Forest Research Institute, Kangding, China
[3] Biogas Institute of Ministry of Agriculture, Chengdu, China

## ABSTRACT

**Background:** The medicinal plants of *Radix et Rhizoma Nardostachyos* include *Nardostachys jatamansi* and *N. chinensis*. Traditionally, the two plants have been used to treat many diseases. Because of their special aroma, they are also commonly used in the food and cosmetics industry. Recently, *N. jatamansi* and *N. chinensis* have been overexploited due to their economic importance, resulting in a sharp decline in their wild resources. Predicting potential distributions of the genus *Nardostachys* under different climate scenarios and understanding its preferred habitat are of great significance for their conservation, artificial cultivation, and assessment of their value.

**Methods:** The Maxent model was used to predict the potential geographical distributions of the genus *Nardostachys* under current and future climatic conditions based on two representative concentration pathways (RCP2.6 and RCP8.5) for the 2050s and 2070s. These data were used to study the effects of climate variables.

**Results:** The results show that the potential distribution of the two species will increase, thus more suitable habitats will be present in China. The suitable habitat for *N. chinensis* presents a relatively stable growth compared to *N. jatamansi*. In addition, precipitation plays a crucial role in modeling the effects of climate change on the genus *Nardostachys*. This study provides theoretical guidance for the cultivation of *N. chinensis*.

Corresponding authors
Wenguo Wang,
wangwenguo@caas.cn
Yang He, heyang@cdutcm.edu.cn

# INTRODUCTION

According to the Chinese Pharmacopoeia, the medicinal plants of *Radix et Rhizoma Nardostachyos* include *Nardostachys jatamansi* and *N. chinensis* (*Pharmacopeia Committee of P. R. China, 2010*). These species are perennial, rhizomatous, in the family Valerianaceae (*Airi et al., 2000*). *N. jatamansi* is about 5–50 cm in height, with a thick and long woody stem. The dark gray main root is thick and long, densely covered with the leaf sheath fibers, with strong fragrance (*Bhat & Malik, 2017*; *Zhang et al., 2016*).

The main difference between *N. jatamansi* and *N. chinensis* is that the obliquely stretched rhizome of the *N. jatamansi*, which with more dense leaf-based fibers to cover the stem, and the corolla tube is wider and shorter (*Flora of China Editorial Committee, 2006*). Also, *N. jatamansi* is mainly found growing between altitudes of 3,300–5,000 m, and it originated to the elevated ranges of the Himalayas in Nepal but also found in the high lands of Sikkim, Bhutan, and Punjab (*Malik, Firdose & Bhat, 2018*). In China, these species were primarily distributed in the eastern part of the Tibetan Plateau and its margins (*Flora of China Editorial Committee, 2006*). Although there are some differences in the morphology and distribution of these two species, they contain similar active ingredients and medical efficacy (*An Editorial Committee of the Administration Bureau of Traditional Chinese Medicine, 2000*). These rhizomes have been used in Chinese traditional medicine. Their medicinal properties include anti-epileptic, anticonvulsant, anti-depressant, anti-inflammatory, sedation, antiarrhythmic, antiparkinson, antibacterial, hypolipidemia, hepatoprotection, and nerve development (*Ahmad et al., 2013*). Recent research showed that the rhizome contains chemical components such as sesquiterpenes, flavonoids, polysaccharides, and lignans (*Chatterjee et al., 2005*). Because of the high content and special aroma of sesquiterpenes, these species have also been used for flavoring (*Le et al., 2017*).

Due to their wide application in medicine, food, cosmetics, and other industries, the market demand for the two species was increasing. However, due to the underdevelopment of the cultivation technology of *Nardostachys*, the industry mainly depended on wild resources. To meet the demands of the market, excessive wild collection has led to a sharp decrease in wild *Nardostachys* populations and serious habitat destruction. Measures need to be taken to protect the species and develop artificial planting technology for them.

The development of effective cultivation and protection of *Nardostachys* species is required to better meet the market demands and sustainably use the species. For long-term and effective development of cultivation of the *Nardostachys* species, it is essential to understand its suitable climate and geographic distribution in future.

Environmental factors have significant influences on the distribution of species. Global climate change is occurring at an unprecedented rate. And the global temperatures are expected to rise significantly in this century, even the greenhouse gas concentrations remain constant at the year 2000 levels (*Rivaes et al., 2013*). Representative concentration pathways (RCPs) were used in climate modeling and research to describe the four possible future climates in the fifth assessment report of the Intergovernmental Panel on Climate Change (*IPCC, 2013*). The four RCPs (RCP2.6, RCP4.5, RCP6, and RCP8.5) are named in the possible range of Radiative Forcing values in the year 2100 relative to pre-industrial values (+2.6, +4.5, +6.0, and +8.5 $W/m^2$, respectively) (*Ma & Sun, 2018*).

Species distribution models could simulate the potential ecological niche of a species from various hypotheses about how environmental factors control the distribution of species and communities based on known occurrence points (*Zhang et al., 2016*). Therefore, the species distribution models could provide the basis for planting land selection and growing climatic conditions during the cultivation of the two plants. The main species distribution models for species distribution predictions include Bioclim,

Climex, Domain, Garp, and Maxent maximum entropy modeling. Among these models, the Maxent model has a strong advantage in predicting species distribution and gained popularity, which is a presence-only modeling technique using a machine learning approach (*Phillips, Anderson & Schapire, 2006*). The Maxent model has successfully completed the prediction of the potential geographical distributions of multiple species for the past few years, including *Fritillaria cirrhosa*, *Lilium nepalense*, *Phellodendron amurense* Rupr, and *Atractylodes lancea* (*Rana et al., 2017*; *Wan et al., 2014*; *Shoudong et al., 2017*; *Zhao et al., 2018*).

*Radix et Rhizoma Nardostachyos* is an endangered and valuable Chinese herbal medicine that needs to be protected and developed. Moreover, its two medical plants with similar medical effects, and more importantly, the two species are on the verge of extinction. For better utilization and development of *Radix et Rhizoma Nardostachyos*, climate change was used as a screening factor to select one species that is more suitable for artificial cultivation. The subject of this paper is to discover the effects of climate change on the two species by the Maxent model, and provide a basis for their artificial cultivation.

## MATERIALS AND METHODS

The complete analysis workflow was summarized in Fig. 1.

### Basic geographic data

A total of 1:400,000 vector map of China's administrative divisions was downloaded from the national basic geographic information system as the base map for the analysis.

### Study area

*N. jatamansi* and *N. chinensis* were mainly distributed in subalpine to alpine areas, where the elevation is more than 3,000 m. The Qinghai–Tibetan Plateau is the main distribution area, ranging from 25° to 40°N, 74° to 104°E. The Qinghai–Tibetan Plateau is the world's highest plateau, with an altitude of between 3,000 and 5,000 m and an average elevation of over 3,500 m. The climate of the Qinghai–Tibetan Plateau is monsoonal. The annual average temperature in most parts of the region is below 8 °C, and the annual precipitation is 400–900 mm/year.

### Data collection

Occurrence records of *N. jatamansi* and *N. chinensi*s were mainly obtained through online databases and consulting literature. The online database includes the Chinese Virtual Herbarium (http://www.cvh.ac.cn/), Specimen Resources Sharing Platform for Education (http://mnh.scu.edu.cn/), and Global Biodiversity Information Facility (https://www.gbif.org/). The literature used was open-source and available worldwide. In the absence of specific geographical coordinate records, Google Earth 7.0 was utilized to find the approximate latitude and longitude according to the described geographical location. After duplicate and invalid records were removed, the sampling points of *N. jatamansi* and *N. chinensi*s were 183 and 60, respectively. We obtained sampling points from the specimen database and literature, and all the specimens collected in the specimen database are wild. In accordance with the requirements of the Maxent software, the distribution

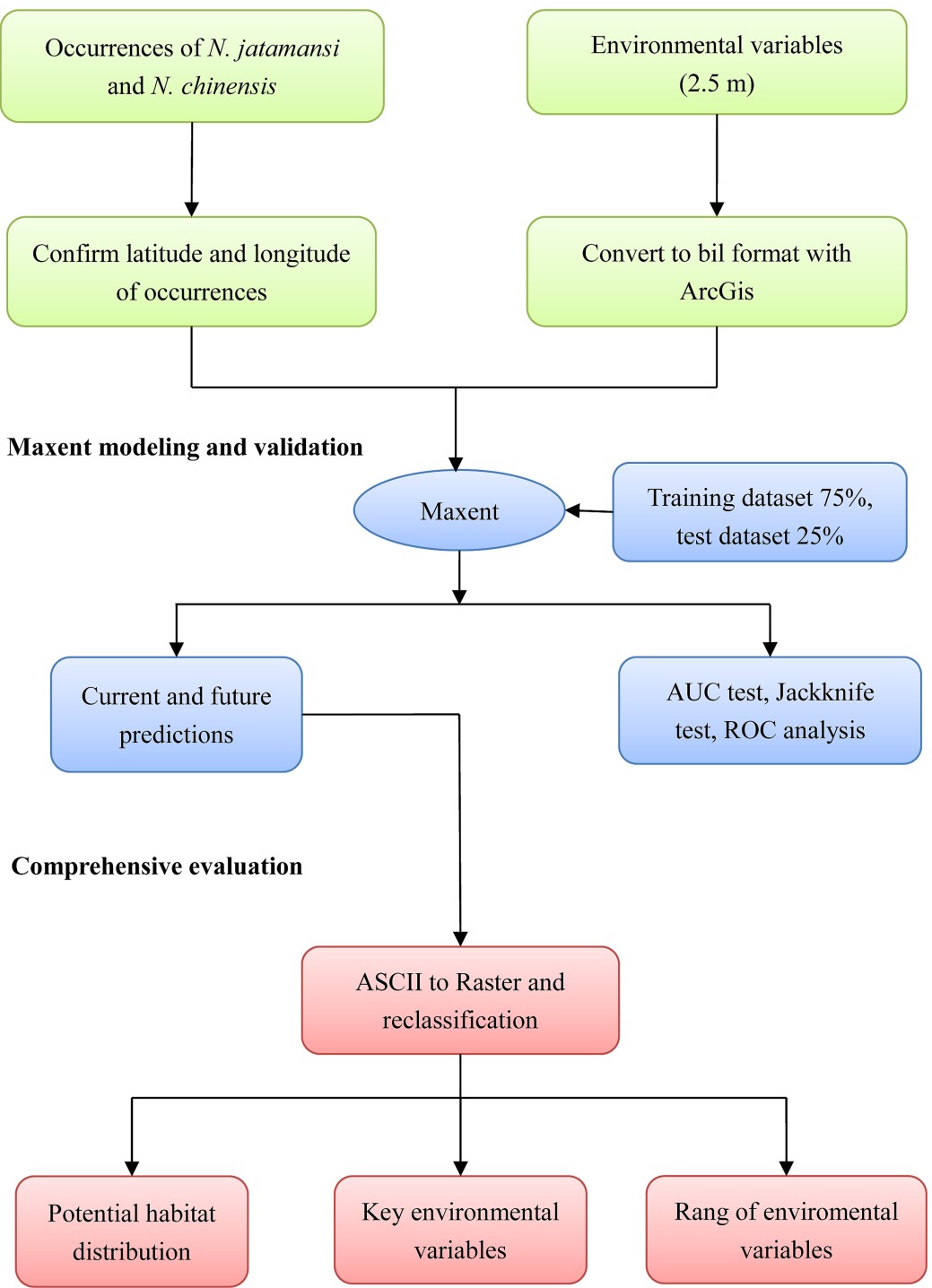

**Figure 1 The processing method in the flowchart.** The study mainly included three processes of data preparation, Maxent simulation, and comprehensive evaluation.

records of *N. jatamansi* and *N. chinensi*s were organized into .csv format files including the species name and distribution point longitude and latitude, with the east and north latitudes positive and the west and south latitudes negative.

## Environmental variables

Climate variables are crucial in simulating the distribution of species. This article mainly analyzed the impact of climate change on the distribution of *N. jatamansi* and *N. chinensi*s. Refer to other literature, 19 bioclimatic variables with 2.5-min resolution of the current conditions and future conditions (2050s and 2070s) were downloaded from the WorldClim-Global Climate Data (http://worldclim.org/) (*Suwannatrai et al., 2017*). The 2.5-min resolution could meet the simulation requirements based on the published article (*Bai, Wei & Li, 2018*; *Qu, Wang & Zhang, 2018*). These variables are derived from monthly temperature (maximum and minimum) and rainfall data, which express seasonality, annual trends and extreme conditions (*Mendoza-Gonzalez et al., 2013*). The 19 bioclimatic variables with strong biological significance explained the adaptation of species with extreme environmental factors. Here, two greenhouse gas concentration scenarios, RCP 2.6 (the minimum greenhouse gas emission scenario) and RCP 8.5 (the maximum greenhouse gas emission scenario) were selected under the CCSM4 (The Community Climate System Model version 4.0) model in this work. Referring to the research of *Ma & Sun (2018)*, RCP2.6 and RCP8.5 are the most common emissions scenarios. The CCSM4 climate model, which was developed by the National Center for Atmospheric Research of the United States, has been thoroughly evaluated in China. It has recently been verified that the CCSM4 model can better simulate the climate characteristics of East Asia (*Tian & Jiang, 2013*).

## Distribution modeling

The Maxent model was a widely used and effective model for species distribution modeling because it produced believable results. It is based on the freely available Maxent software V3.3.3k (https://github.com/mrmaxent/Maxent/blob/master/ArchivedReleases/3.3.3k/maxent.jar). Maxent was used to predict the current and future potential distributions of *N. jatamansi* and *N. chinensi*s based on the occurrence records and climate data. The logical output (results are expressed in terms of probability) was selected to assess the likelihood of the presence of a species in each grid cell. A total of 25% of the occurrence records were chosen as the test set, and the remaining parameters were the default settings (*Yi et al., 2016*). Jackknifing in the environment parameter settings was checked to measure the percentage contribution of each environmental variable and to evaluate the importance of environmental factors in affecting the geographical distribution of species.

The area under the receiver operating curve, called AUC, was used to evaluate the accuracy of the prediction results (*Zhang et al., 2016*). AUC values range from zero to one, and the larger the AUC value, the farther away from the random distribution. High AUC values indicate the better prediction of the model. AUC values generally range from 0.5 to 0.7 when accuracy is low; moderate accuracy is generally 0.7~0.9; a high degree of accuracy is typically greater than 0.9 (*Wakie et al., 2014*).
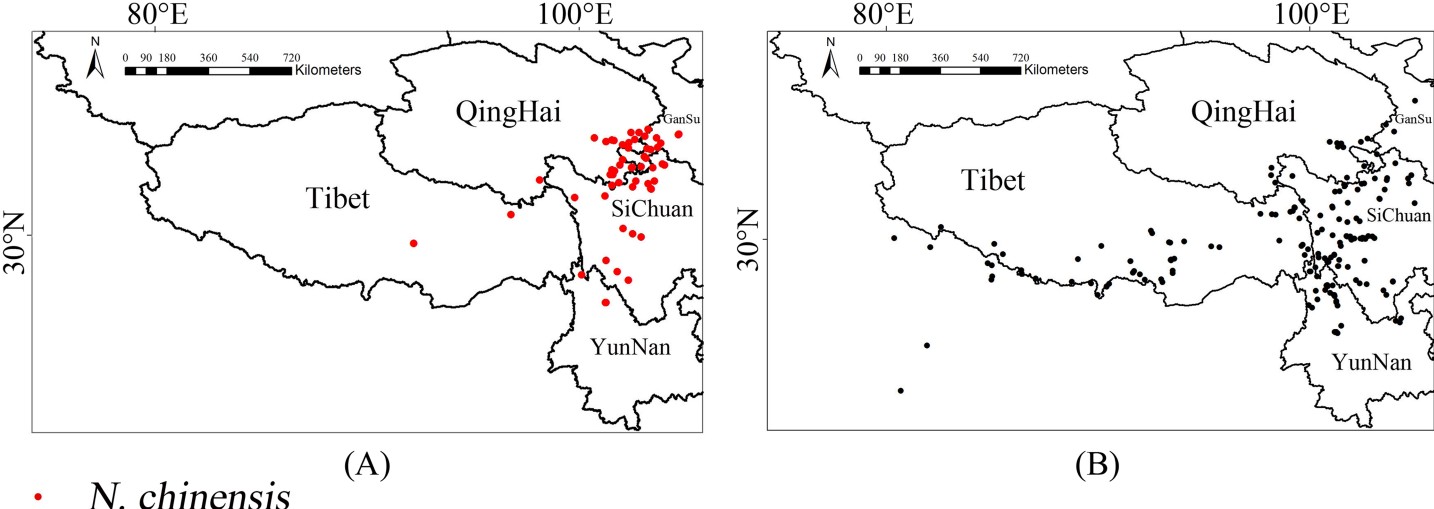

**Figure 2 The original occurrence records.** Red dots represented georeferenced locations of 60 occurrences of *N. chinensis* (A), and black dots represented georeferenced locations of 183 occurrences of *N. chinensis* in Qinghai–Tibetan Plateau (B).

## RESULTS

### Occurrences of *N. jatamansi* and *N. chinensis*

Figure 2 showed the original occurrences of *N. jatamansi* and *N. chinensi*s were mainly distributed in Sichuan Province, parts of Yunnan, Gansu, and Qinghai Provinces, and the Tibetan Autonomous Region. The main areas in which *N. jatamansi* was distributed were the Tibetan Autonomous Region, Sichuan Province, Yunnan Province, and a small part of Qinghai and Gansu Provinces. In addition, *N. jatamansi* was also distributed in other countries including India and Nepal. Hence, the distribution of *N. jatamansi* is more extensive than that of *N. chinensis*.

### Predicting the distribution under current conditions

Figures 3A and 3F showed the predicted area of distribution for *N. jatamansi* and *N. chinensi*s by the Maxent model under the current climate conditions. The simulation output results from the Maxent software are between zero and one. According to a previously published article by *Wang et al. (2014)*, the results were divided into four levels: 0–0.05, 0.05–0.25, 0.25–0.5, and 0.5–1. *Elith et al. (2011)* considered that the average probability of species being present in suitable areas was 0.5. For the two species in this study, the area with probabilities above 0.5 was the main natural distribution area. Thus, the probabilities above 0.5 was defined as suitable habitats, and probabilities less than 0.5 were unsuitable based on the study of *Wang et al. (2018)*.

As showed in Figs. 3A and 3F, suitable areas under current conditions of *N. jatamansi* were mainly distributed in the Qinghai–Tibetan Plateau in China, including western parts of Sichuan Province, northern Yunnan Province, the eastern part of the Tibetan Autonomous Region, and a small part of southern Qinghai and Gansu Provinces.
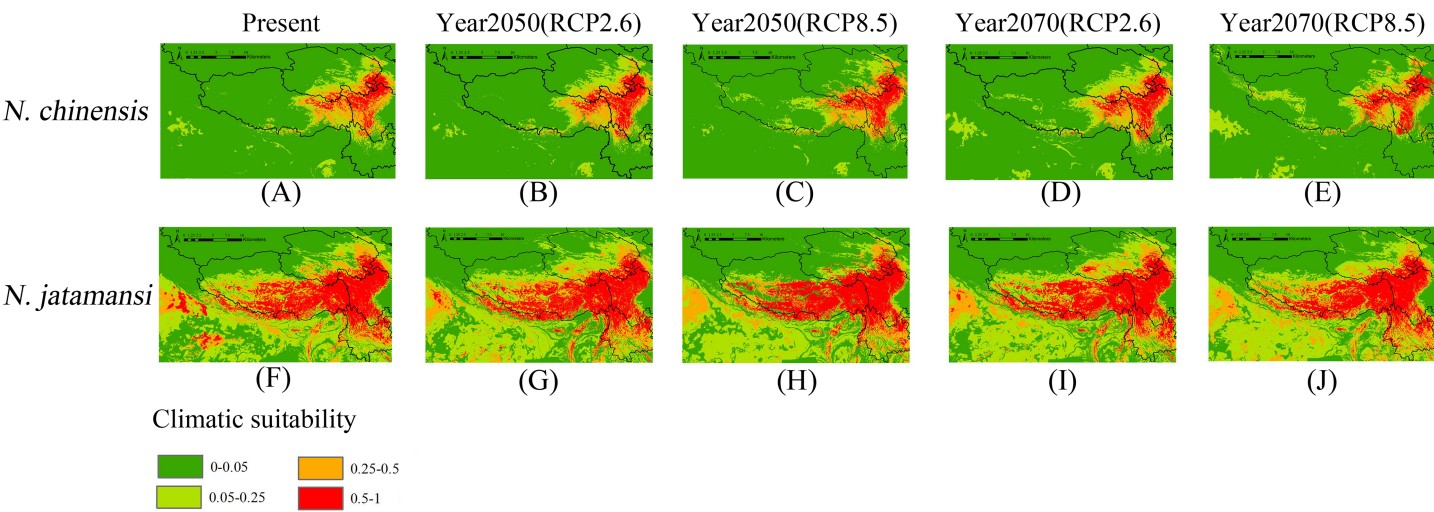

| Present | Year2050(RCP2.6) | Year2050(RCP8.5) | Year2070(RCP2.6) | Year2070(RCP8.5) |

*N. chinensis*

(A)  (B)  (C)  (D)  (E)

*N. jatamansi*

(F)  (G)  (H)  (I)  (J)

Climatic suitability

- 0-0.05
- 0.05-0.25
- 0.25-0.5
- 0.5-1

**Figure 3** **Present (A and F), and future (B–E and G–J) potential spatial distribution of *N. jatamansi* and *N. chinensis*.** Future predictions (2050s and 2070s) were based on two representative concentration pathways under CCSM4 climate model.

In addition, it has a small suitable area in India, Bhutan, and Nepal. Among these distributions, the suitable areas of Sichuan, Tibet, and Yunnan are extensive and concentrated in the high mountain plateau region, which was roughly in line with the records of the distribution of *N. jatamansi*.

Suitable areas of *N. chinensis* were mainly distributed at the junction of the Tibetan Autonomous Region and Sichuan and Qinghai Provinces in China. Sichuan Province has the most suitable habitats. This was consistent with the collection records from Sichuan, indicating that *N. chinensis* was primarily distributed in the province, and no suitable habitats were indicated in other countries. According to the prediction, the area in which *N. jatamansi* is distributed was significantly larger than the distribution of *N. chinensis* under current conditions.

### The effects of climate change on the geographical distribution of *N. jatamansi* and *N. chinenis*

Figures 3B–3E and 3G–3J showed the potential geographical distributions of *N. jatamansi* and *N. chinenis* predicted by the Maxent model under the CCSM4 of the 2050s and 2070s. There was an overall upward trend in the area of suitable habitats for *N. jatamansi* and *N. chinensis* (Fig. 4).

By 2050, suitable areas (RCP2.6 and RCP8.5) of *N. jatamansi* increased. Relative to the current situation, both ranges increased in RCP2.6 and RCP8.5 with a greater growth rate under RCP8.5. The scope of suitable areas in Qinghai and Tibet increased obviously and exhibited a trend of northward expansion. Suitable areas of *N. chinensis* under RCP2.6 and RCP8.5 emissions scenarios also increased compared to the current situation. The scope of suitable areas in western Sichuan Province and Tibet increased obviously and also exhibited a trend of northward expansion.

By 2070, the area of suitable habitats for *N. jatamansi* under the RCP2.6 emission scenario will continue to grow. However, the range of suitable areas under the RCP8.5

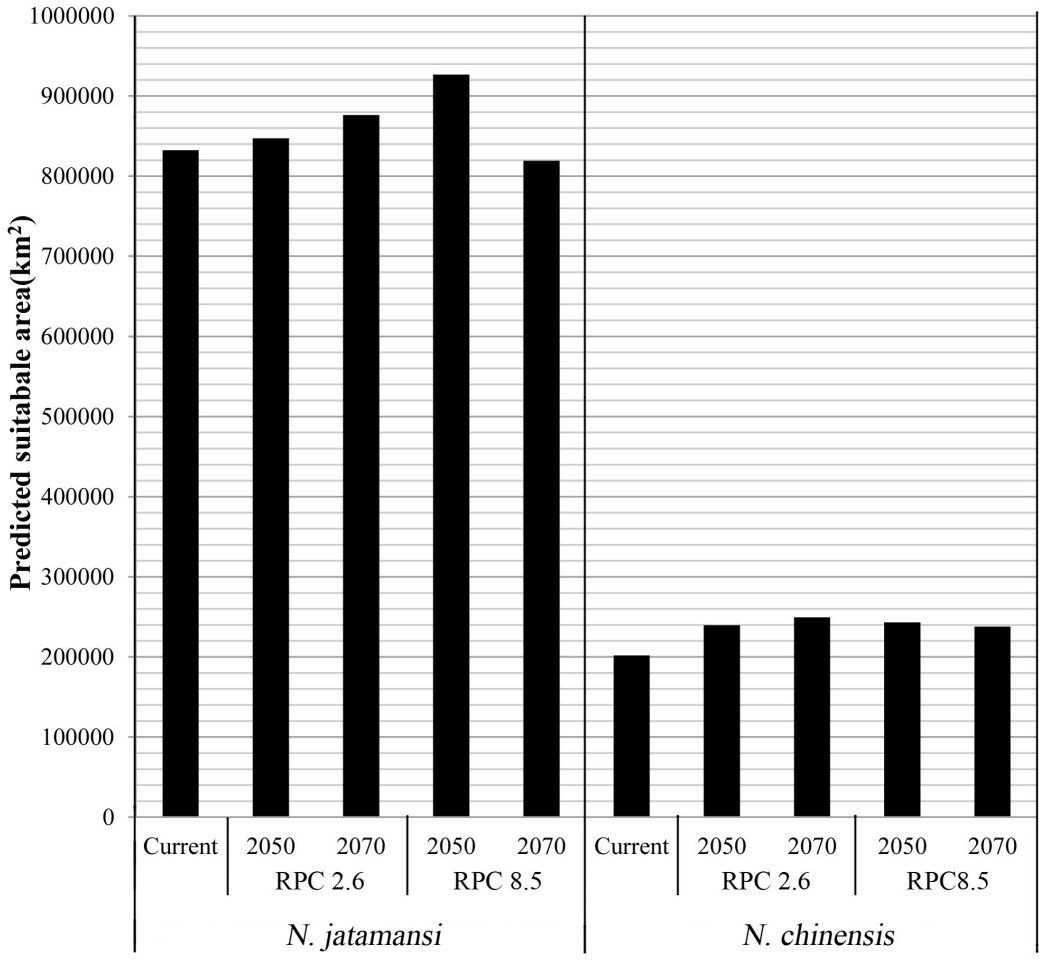

**Figure 4 The area of suitable habitats of *N. jatamansi* and *N. chinensis* for present and future conditions predicted by Maxent based on CCSM4 climate model.** There was an overall upward trend in the area of suitable habitats for *N. jatamansi* and *N. chinensis*.

emission scenario was significantly reduced and was significantly lower than the current level. The results showed that the climate-suitable area decreased in the 2070s under the RCP8.5 climate scenario compared to the current period.

Suitable areas of *N. chinensis* under RCP2.6 also continued to increase, relative to the 2050s and the present. Suitable areas of *N. chinensis* under RCP8.5 still increased compared to the present. However, suitable areas of *N. chinensis* under RCP8.5 decreased from the 2050s.

## Contribution of environmental variables

A total of 19 bioclimatic variables from the BIOCLIM climate dataset were used as the background environmental data, such as temperature and rainfall. These variables are comprehensive conclusions and quantitative descriptions of the climatic environment in the regions containing the targeted species. The potential distributions of *N. jatamansi* and *N. chinensis* were simulated by the Maxent model, which automatically predicted

**Table 1 Bioclimatic variables used for model construction and their percentage contribution in predicting the current distribution of N. jatamansi and N. chinensis using MaxEnt.**

| Abbreviation | Description | Percent contribution | |
|---|---|---|---|
| | | N. jatamansi | N. chinensis |
| Bio1 | Annual mean temperature (°C) | 0.8 | 1.5 |
| Bio2 | Mean diurnal range (mean of monthly (max temp–min temp)) (°C) | 0.1 | 0 |
| Bio3 | Isothermally (Bio2/Bio7) (×100) | 15.4 | 7.7 |
| Bio4 | Temperature seasonality (standard deviation × 100) (C of V) | 0.1 | 0 |
| Bio5 | Max temperature of warmest month (°C) | 5.2 | 8.1 |
| Bio6 | Min temperature of coldest month (°C) | 0.5 | 3.4 |
| Bio7 | Temperature annual range (Bio5–Bio6) (°C) | 0.3 | 0.3 |
| Bio8 | Mean temperature of wettest quarter (°C) | 9.6 | 9.6 |
| Bio9 | Mean temperature of driest quarter (°C) | 0.3 | 0 |
| Bio10 | Mean temperature of warmest quarter (°C) | 5.3 | 2.7 |
| Bio11 | Mean temperature of coldest quarter (°C) | 0.2 | 0 |
| Bio12 | Annual precipitation (mm) | 0.2 | 0.1 |
| Bio13 | Precipitation of wettest month (mm) | 13 | 15.2 |
| Bio14 | Precipitation of driest month (mm) | 36 | 36 |
| Bio15 | Precipitation seasonality (C of V) | 0.6 | 0.1 |
| Bio16 | Precipitation of wettest quarter (mm) | 0.1 | 2.5 |
| Bio17 | Precipitation of driest quarter (mm) | 8.6 | 0.2 |
| Bio18 | Precipitation of warmest quarter (mm) | 0.5 | 0.1 |
| Bio19 | Precipitation of coldest quarter (mm) | 3.4 | 12.6 |

the contribution percentage of each variable to the forecasted results. Table 1 listed selected bioclimatic variables and their percent contribution in the Maxent model for *N. jatamansi* and *N. chinensis* in their current distribution range. Among them, precipitation of the driest period (Bio14) was the largest contributing factor when modeling the distributions of *N. jatamansi* and *N. chinensis*.

The jackknife test also showed that precipitation of the coldest quarter (Bio19), precipitation of the driest quarter (Bio17), and precipitation of the driest period (Bio14) were the main variables in the simulation of *N. chinensis* (Fig. 5). However, isothermality (Bio3), precipitation seasonality (Bio15), and precipitation of the driest period (Bio14) were the main factors affecting the modeled distribution of *N. jatamansi* (Fig. 5).

Figure 6 showed the response curves for single climate variables in the Maxent forecast of *N. jatamansi* and *N. chinensis*. Response curves indicated the correlation between the environmental variables used for prediction and the probability of the presence of *N. jatamansi* and *N. chinensis*. In Fig. 6, Bio14 response curves peak in the distribution predictions of *N. jatamansi* and *N. chinensis*. The distribution probability of *N. jatamansi* and *N. chinensis* increased with the increase of variable value. After exceeding this peak, the distribution probability decreased with the increase of the value for environmental factors, indicating that *N. jatamansi* and *N. chinensis* have some adaptability to these

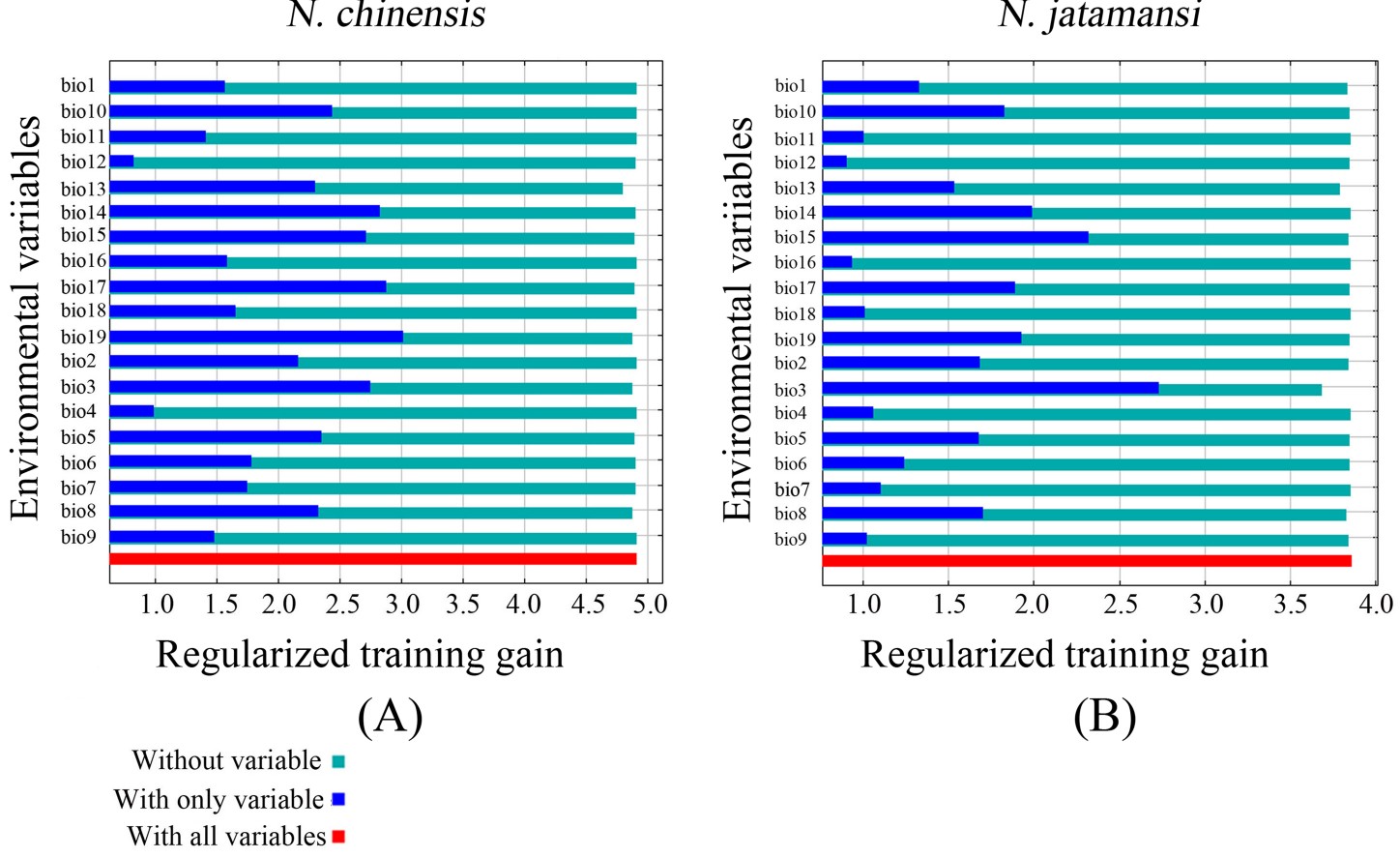

**Figure 5 The Jackknife test for evaluating the relative importance of environmental variables for *N. chinensis* (A) and *N. jatamansi* (B).** The blue, aqua, and red bars represent the results of the model created with each individual, all remaining variables, and all the variables, respectively.

variables. When the probability is greater than 0.5, the corresponding ecological factor value is more suitable for the growth of the plant.

The prediction showed that the best ranges for these indicators were precipitation of the driest month, one to five mm for *N. chinensis* and 0–10 mm for *N. jatamansi*. The optimum range of precipitation of the driest month is very narrow, indicating that *N. jatamansi* and *N. chinensis* were sensitive to precipitation.

## DISCUSSION

With global warming, upon changes in temperature, precipitation, and $CO_2$ concentration, and the changes in certain climatic resources changed the dynamics and balance of different species, thereby affecting their productivity (*Rivaes et al., 2014*). When the distributions of climate resources adapting to *N. jatamansi* and *N. chinensis* change, the geographical distributions also undergo corresponding changes. The temperature increases strongly under RCP8.5 and may exceed the suitable temperature for plant growth (*Lv & Zhou, 2018*). Thus, the climate-suitable area decreased in the 2070s under the RCP8.5 climate scenario compared to the current period.
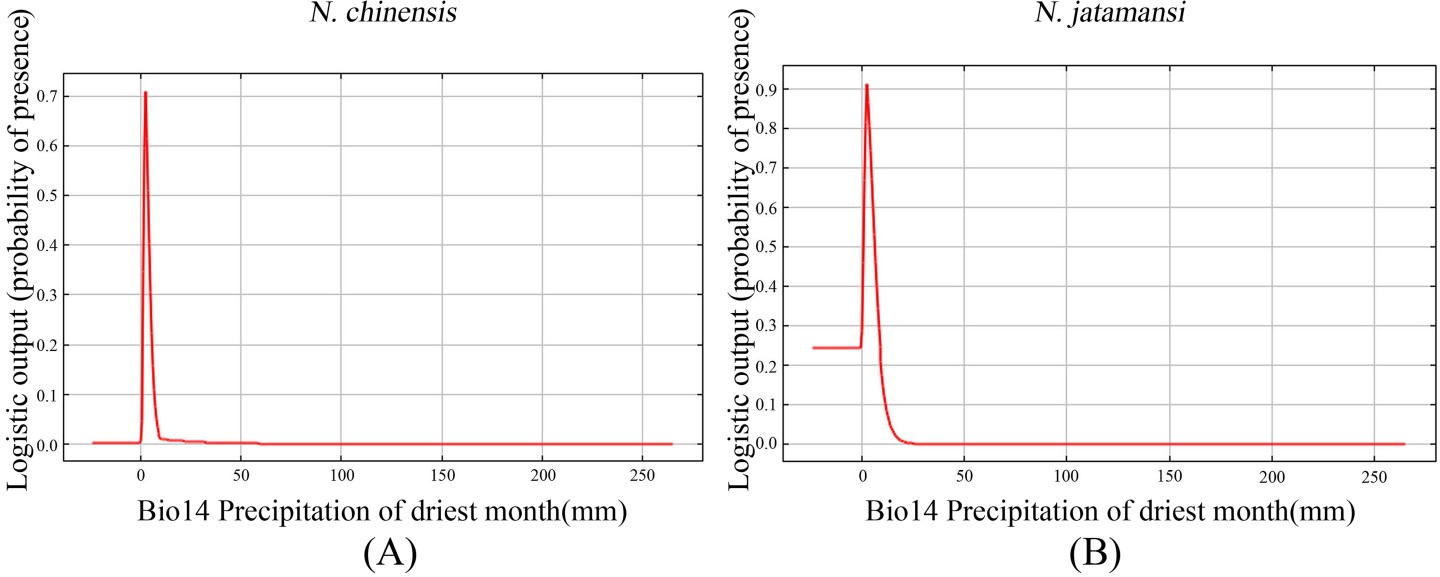

**Figure 6** **The response curves of precipitation of driest month under the current period for *N. chinensis* (A) and *N. jatamansi* (B).** Response curves indicated the correlation between the environmental variables used for prediction and the probability of the presence of *N. jatamansi* and *N. chinensis*.

This indicated that more areas will be suitable for the distribution of *N. jatamansi*, which would have greater potential for expansion in China in the future under a low-emission scenario, contrary to high-emission scenarios. In addition, suitable areas in Qinghai and Tibet decreased obviously and exhibited a trend of northward expansion. Nevertheless, suitable sites in the high altitude area of Tibet markedly increased. And previous studies have shown that with global warming, some species will migrate to high latitude or high elevation (*Zhang et al., 2018*). The variation of the suitable area of *N. jatamansi* was consistent with this result.

*Ma & Sun (2018)* had predicted the distribution of *Stipa purpurea* across the Tibetan Plateau via the Maxent model. The habitat suitability of *S. purpurea* in RCP2.6 and RCP8.5 revealed that the habitat suitability for the species increased from the 1990s to 2050s, then decreased from the 2050s to 2070s. Their results indicated the trend of the suitable habitat area was consistent with our prediction. However, they also considered the impact of topographic variables in the study, which was our shortcoming.

According to the above results, the suitable habitat *N. jatamansi* varies greatly under different climate scenarios and was more affected by climate change. However, the potential distributions of *N. chinensis* showed a growing trend in different climate scenarios compared to its current situation and was less affected by climate change. Therefore, in order to avoid the impact of climate change on the artificial cultivation of the genus *Nardostachys,* and achieve stable development, we could choose *N. chinensis* as the main cultivated species.

Studying the interaction between species and the environment is an essential aspect of species cultivation. The relationship between the probability of species presence and dominant environmental variables was analyzed in this paper. The percent

contribution of bioclimatic variables, the jackknife test and the response curves were created by the MaxEnt software.

According to the results of above forecase, we can conclude that precipitation of the driest period is the most important environmental factor affecting the growth of *N. chinensis* and *N. jatamansi*. The research has shown that while comparing the distribution of the two contrasting hill slopes (east and west), the density of *N. jatamansi* has increased significantly with altitude was revealed on a west-facing slope. And the north and west-facing slopes in the alpine zone of this region are broadly considered as the shady areas with low light and high moisture (*Airi et al., 2000*). The advantage of high water and nutrient supply for rhizome and root production has also been reported elsewhere for related *valerian* species (*Bernath, 1997*). These results indicated that the precipitation plays a dominant role in simulating the effects of climate change on *N. jatamansi* and *N. chinensis* compared to temperature and other environmental variables.

The analysis for the response curves showed that the probability of species presence changed as a result of the dominant environmental variables. The results indicated that *N. jatamansi* and *N. chinensis* were sensitive to precipitation. Thus, this showed that research on water management should be strengthened for future cultivation practices.

## CONCLUSION

The results of this Maxent simulation of climate change impacts on the distribution of two *Nardostachys* species and prediction of their potential distributions under current and future climate conditions are reliable. Considering the effects of climate change, *N. chinensis* is more suitable for cultivation. In addition, precipitation of the driest period is an important factor in the impact of climate change on *N. jatamansi* and *N. chinensis*. It is of great significance to study the impact of climate change on the genus *Nardostachys* to predict their potential distribution in the future for the protection, sustainable use, and value assessment of these biological resources. It is essential to conduct future studies on the effect of watering of *N. chinensis* for developing effective techniques to cultivate *N. chinensis*. The study will provide references for the cultivation planning and resource utilization of *N. jatamansi* and *N. chinensis*.

### Funding
This work was supported by the Sichuan Science and Technology Project (Grant No. 2018FZ0048). The funders had no role in study design, data collection and analysis, decision to publish, or preparation of the manuscript.

### Grant Disclosures
The following grant information was disclosed by the authors:
Sichuan Science and Technology Project: 2018FZ0048.

### Competing Interests
The authors declare that they have no competing interests.

## Author Contributions

- Junjun Li conceived and designed the experiments, performed the experiments, analyzed the data, prepared figures, and/or tables, authored or reviewed drafts of the paper, approved the final draft.
- Jie Wu performed the experiments, analyzed the data, contributed reagents/materials/analysis tools, approved the final draft.
- Kezhong Peng performed the experiments, approved the final draft.
- Gang Fan performed the experiments, analyzed the data, approved the final draft.
- Haiqing Yu performed the experiments, approved the final draft.
- Wenguo Wang conceived and designed the experiments, analyzed the data, contributed reagents/materials/analysis tools, prepared figures, and/or tables, authored or reviewed drafts of the paper, approved the final draft.
- Yang He conceived and designed the experiments, analyzed the data, contributed reagents/materials/analysis tools, prepared figures, and/or tables, authored or reviewed drafts of the paper, approved the final draft.

## Data Availability

The distribution and bioclimatic raw data of genus Nardostachys are available at FigShare: He, Yang (2018): Bioclimatic variables.zip. figshare. Fileset. https://doi.org/10.6084/m9.figshare.7304393.v1.

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
