# Peer review of "Simulating the effects of climate change across the geographical distribution of two medicinal plants in the genus Nardostachys"

_PeerJ, doi:10.7717/peerj.6730_

## Round 0.1 · original submission · Major Revisions

Thank you for your patience in this process and apologies for delays from the holiday season. I have received two reviews on this manuscript. One is highly critical (reject) and one requests major revisions. I am inclined to agree with Major Revisions, but caution the revision to directly address the comments of both reviewers. The rejection review is quite detailed and provides good feedback on how to improve the manuscript. Please work through both of these reviews and resubmit.

Thank you.

Reviewer 1 ·

Basic reporting

1. The English language should be improved to ensure that readers can clearly understand your text. Clear and professional English is not used. For examples, "RPC" should be "RCP".
2. The introduction and background descriptions are too simple to introduce the biological characteristics of species and the current research status of SDMs, nor to provide relevant references. Such as, "Species distribution models can help solve this problem. The main ecological niche models for species distribution predictions include Bioclim, Climex, Domain, Garp, and MaxEnt maximum entropy model."
3. Figure 1 has no latitude and longitude. The label of Figure 2 has some problems. The pictures and texts do not correspond.
4. The frame structure of the article is unreasonable. For example, the relevant description of RCPs should be put in the introduction.

Experimental design

1. " After duplicate and invalid records were removed, a total of 183 records were collected. " How many sampling points are there for the two species respectively? How to identify them as wild species?
2. " Nineteen bioclimatic variables with 2.5-minute resolution of the current conditions and future conditions... " Why not use data with a resolution of 30 seconds? The environmental data is not filtered to reduce the correlation between the data. Why not consider topography, soil, light and other factors?
3. “ the results were divided into four levels using ArcGis 10.2: 0-0.05, 0.05-0.25, 0.25-0.5, and 0.5-1. ” What is the basis for the results of suitability classification? Why are the first three categories defined as unsuitable areas?
4. " Representative Concentration Pathways (RPCs) 2.6 and 8.5, were selected..." Please explian why RCP 2.6 and RCP 8.5 were chosen.
5. From the aspects of research content, model construction and research results, there is no innovation in this paper.

Validity of the findings

1. Comparing the impacts of climate change on the area of suitable habitats , the specific scope and reasons of the area change are not explained.
2. "Therefore, developing artificial cultivation techniques for N. cdinensis is more beneficial from a long-term prospective." The reason for choosing N. cdinensis for artificial cultivation is illogical.
3. " precipitation is an important factor in the impact of climate change on N. jatamansi and N. cdinensis." The results show that the effect of precipitation on species distribution is greater than that of temperature, but the basis is insufficient, which needs further explanation.

Reviewer 2 ·

Basic reporting

The authors have presented a good background on this topic. The language use is professional, and the paper is structured conform journal guideline. Please increase the size of the text in Figure 1, 2, 4 and 5.

Experimental design

The manuscript is within the journal’s scope. It has well-defined research questions considering the current knowledge gap in the literature, followed by a rigorous investigation. Methods are well described however I would recommend explaining the method through the flowchart but in a more symbolic way, not just text.

Validity of the findings

Data and statistical analysis are robust and controlled.

Additional comments

Authors of this manuscript have presented a study concerning the impact of the effects of climate change across the geographical distribution of two medicinal plants in the genus Nardostachys. The author has applied entropy models MaxEnt to predict the potential geographical distributions of the genus Nardostachys under current and future climatic conditions based on two representative concentration pathways (RPC2.6 and RPC8.5) for the 2050s and 2070s. The authors have found that the potential distribution of the two species will increase, thus more suitable habitats will be present in China. The topic is interesting, an important issue and generally well written, well structured and contributes to the existing knowledge. However, there are still some occasional grammar errors through the manuscript especially the article ‘’the’’, ‘’a’’ and ‘’an’’ is missing in many places, please make a spellchecking.

The results and discussion section needs further improvement, compare your findings with the other author's conclusions.
• In general, the manuscript needs to shorten, there irrelevant and redundant information in the text.
• Please provide more deep discussion about your results, compare your findings with the other author findings.
• Please clearly state the novelty of this work.
• Please check the reference style, some of the references are not according to the journal style, especially the journals abbreviations.
• At present, neither the original contributions of the paper nor its practical significance is clear for me.
• As said, it could be a failure to comprehend the main argument; and lightening the discussion would enhance comprehensibility and make the authors’ point clearer and stronger.
• Therefore, the reviewer recommends to further improve the manuscript before accepting it for publication. Some of the specific comments are listed below.
• Please consider citing following literature; reviewer believes that it may help the authors to have a more holistic overview about climate impact on plants:

Rivaes, R., Rodríguez‐González, P. M., Albuquerque, A., Pinheiro, A. N., Egger, G., & Ferreira, M. T. (2013). Riparian vegetation responses to altered flow regimes driven by climate change in Mediterranean rivers. Ecohydrology, 6(3), 413-424.
Lv, X., & Zhou, G. (2018). Climatic Suitability of the Geographic Distribution of Stipa breviflora in Chinese Temperate Grassland under Climate Change. Sustainability, 10(10), 3767.
Rivaes, R. P., Rodríguez-González, P. M., Ferreira, M. T., Pinheiro, A. N., Politti, E., Egger, G.,& Francés, F. (2014). Modeling the evolution of riparian woodlands facing climate change in three European rivers with contrasting flow regimes. PloS one, 9(10), e110200.

---

## Round 0.2 · accepted · Accept

Congratulations. You have made a full faith effort to revise this work and the paper is much stronger. Thank you for the updated manuscript. Nice work.

#